# Dicarbonyl Stress in Diabetic Vascular Disease

**DOI:** 10.3390/ijms23116186

**Published:** 2022-05-31

**Authors:** Bernd Stratmann

**Affiliations:** Herz- und Diabeteszentrum NRW, Diabeteszentrum, Ruhr Universität Bochum, 32545 Bad Oeynhausen, Germany; bstratmann@hdz-nrw.de; Tel.: +49-5731-97-3768; Fax: +49-5731-97-2410

**Keywords:** methylglyoxal, advanced glycation end products, glycotoxic load, endothelial dysfunction, cardiovascular risk, atherosclerotic disease

## Abstract

Late vascular complications play a prominent role in the diabetes-induced increase in morbidity and mortality. Diabetes mellitus is recognised as a risk factor driving atherosclerosis and cardiovascular mortality; even after the normalisation of blood glucose concentration, the event risk is amplified—an effect called “glycolytic memory”. The hallmark of this glycolytic memory and diabetic pathology are advanced glycation end products (AGEs) and reactive glucose metabolites such as methylglyoxal (MGO), a highly reactive dicarbonyl compound derived mainly from glycolysis. MGO and AGEs have an impact on vascular and organ structure and function, contributing to organ damage. As MGO is not only associated with hyperglycaemia in diabetes but also with other risk factors for diabetic vascular complications such as obesity, dyslipidaemia and hypertension, MGO is identified as a major player in the development of vascular complications in diabetes both on micro- as well as macrovascular level. In diabetes mellitus, the detoxifying system for MGO, the glyoxalase system, is diminished, accounting for the increased MGO concentration and glycotoxic load. This overview will summarise current knowledge on the effect of MGO and AGEs on vascular function.

## 1. Clinical Situation

Diabetes mellitus (DM) is characterised by diminished insulin availability, action, or both. Currently, it is the largest global health problem, irrespective of developmental status. Given the fact that diabetes contributes to increased vascular event risk irrespectively of the type of the disease, we will face an upcoming pandemic with the detected cardiovascular event being just the tip of the iceberg.

Even after the normalisation of blood glucose levels, the increased risk remains, which has led to the assumption of a glycolytic memory (“legacy effect”) contributing to elevated morbidity and mortality in the diabetic population [1]. People with diabetes face micro- and macro- vascular dysfunction, affecting small vessels such as in the retina, the kidney, and the nerves accounting for an elevated risk of 4–20 for morbidity, or larger vessels such as in the coronary system, the carotids, and the peripheral arteries resulting in a two–four-fold risk elevation [2]. The atherosclerosis manifested by the large vessel diseases affecting the coronary arteries (coronary artery disease, CAD), the cerebrovascular system (cerebrovascular disease, CBVD), or the peripheral arteries (peripheral artery disease, PAD) mainly account for the increased mortality risk. In addition to neuropathy, PAD is responsible for the increased risk of diabetic foot syndrome. In terms of cardiovascular mortality and morbidity, heart failure is a major contributor. Advanced glycation (AGEs) and reactive glucose metabolites have been proven to be involved in these pathologies in both types of diabetes mellitus (T1DM and T2DM).

Glycolytic overload causes an extreme accumulation of highly reactive dicarbonyl compounds, resulting in so-called dicarbonyl stress [3]. Methylglyoxal (MGO) is the most relevant dicarbonyl substance in diabetic conditions, triggering non-enzymatic glycation and leading to the irreversible overproduction of AGEs. A hundred years ago, Louis Camille Maillard recognised the browning reaction in which glucose reacts with free amino acid functions in proteins resulting in the formation of AGEs. These can be formed during food processing or intracellularly by an MGO or glucose reaction. Patients with either type of diabetes mellitus present with elevated levels of MGO, and hyperglycaemia is the main cause of MGO (Figure 1) [4]. The glyoxalase system is the main detoxifying system for MGO through the formation of D-lactate [5]. This system is active in the cytoplasm of all mammalian cells and comprises two major enzymes, the glyoxalases, GLO1 and GLO2 [5]. The major substrate for GLO1 is hemithioacetal, which is formed in the first step from MGO and reduced glutathione (GSH). GLO1 catalyses the conversion of the hemithioacetal to S-D-lactoylglutathione. In the next step, glyoxalase 2 (GLO2) catalyses the hydrolysis of S-D-lactoylglutathione, forming D-lactate and recycling GSH. The GLO1-reaction represents the rate-limiting step for the detoxification of MGO [5]. Additionally, in the glyoxalase system, aldehyde dehydrogenases (ALDHs) and aldose reductases (AKRs) may contribute, but to a much lesser extent, to MGO detoxification.

Interestingly, GLO1 expression is under the positive control of the ARE-(stress)-responsive element by the transcription factor, nuclear factor erythroid 2-related factor 2 (Nrf2) [6]. Negative stimulation of the expression occurs via hypoxia-inducible factor 1α (HIF-1α) binding to ARE under hypoxic conditions, accounting for hypoxia-induced MGO-stress [7]. NF-κB antagonises the transcriptional activity of Nrf2; therefore, hyperglycaemia-driven or an inflammation-associated increase in NF-κB accounts for the reduced GLO1 expression [8]. This actually gives rise to therapeutic routes, including Nrf2 activators, to increase GLO1 concentrations.

**Figure 1 ijms-23-06186-f001:**
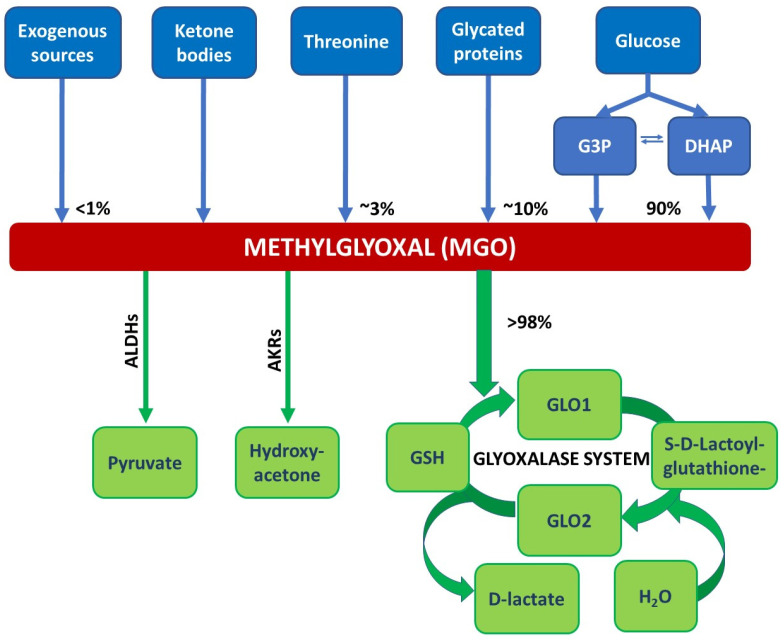
Reactions/sources that lead to the formation and degradation of MGO. MGO is formed via the spontaneous degradation of the triosephosphates dihydroxyacetone phosphate (DHAP) and glyceraldehyd 3-phosphate (G3P) originating from glycolysis. Other sources of MGO are exogenous sources like food and beverages, glycated proteins, threonine, or ketone bodies (acetoacetate). The glyoxalase system detoxifies MGO, in the first step, MGO forms a hemithioacetal with reduced glutathione (GSH). This serves as a substrate for glyoxalase 1 (GLO1). GLO1 catalyses the conversion of the hemithioacetal to S-D-lactoylglutathione. In the next step, glyoxalase 2 (GLO2) catalyses the hydrolysis of S-D-lactoylglutathione, forming D-lactate and recycling GSH. Minor degradation pathways of MGO involve aldehyde dehydrogenases (ALDHs) and aldose reductases (AKRs) yielding in pyruvate or hydroxyacetone, respectively; adapted from [9].

## 2. Endothelial Function

The healthy endothelium forms the innermost layer of the blood and lymphatic vessels and is best described as a multifunctional organ with systemic and tissue-specific roles. It regulates oxygen and nutrient supply, immune cell trafficking, and inflammatory processes. It organises haemostasis and coagulation, vasomotor tone, blood vessel permeability, and angiogenesis. It regulates organ size and function, and it is involved in the development of myocardial hypertrophy in the myocardium. The endothelium might have an impact on liver size and function and is also involved in kidney function [10].

Therefore, the endothelium has to be regarded as a living organ whose function has to be preserved.

However, endothelial dysfunction is provoked by a plethora of factors, with obesity and diabetes mellitus playing leading roles. Endothelial dysfunction is characterised by elevated vasoconstriction, increased permeability, and proinflammatory and prothrombotic states [11]. By hyperlipidaemia, the formation of atherosclerotic plaques is provoked, with the oxidation of low-density lipoprotein (LDL-) cholesterol as a hallmark reaction. On the other hand, the liver faces free-fatty-acid overflow, resulting in inflammation, as can be seen by the elevation of C-reactive protein (CRP) and in the activation of the coagulation system, indicated by increased fibrinogen, plasminogen activator inhibitor type 1 (PAI-1), and thrombosis [12].

Diabetes mellitus induces endothelial dysfunction via the generation of reactive oxygen species (ROS), either derived from hyperglycaemia or glycation, but AGEs and reactive carbonyl species (RCS) exert direct effects on the endothelium as well. Through the increased glycolytic flux induced by hyperglycaemia, glycolysis-derived RCS are generated in addition to hyperlipidaemia, with or without concomitant hyperglycaemia-generate substrates or stress-derived RCS. The reduced function of the detoxifying systems contributes to increased levels of RCS, which may directly induce tissue damage or—via AGE generation or coupling to the AGE-receptor RAGE—switch on the nuclear factor ‘kappa-light-chain-enhancer’ of the activated B-cells (NFκB) system, inducing indirect tissue damage. Reactive oxygen species (ROS) generated by these pathways contribute to lipid peroxidation and further produce substances involved in the deleterious atherosclerotic process [12].

## 3. The Observations—Cells, Animals, Humans

Several research groups have been working on endothelial cells to enlighten the process involved in diabetes- or RCS-induced endothelial dysfunction. In human aortic cells, endothelial function parameters, such as vascular cell adhesion molecule 1 (VCAM-1) and intercellular adhesion molecule 1 (ICAM-1), were altered following the siRNA-induced downregulation of GLO1 under hyperglycaemic conditions. GLO1 is the central enzyme in MGO detoxification. Increased levels of monocyte chemoattractant protein 1 (MCP-1) indicated the induction of inflammatory processes contributing to endothelial dysfunction [13]. Arterial stiffening processes were analysed by collagen-expression profiling, and collagens 1 and 5 were proven to be upregulated. No effect of MGO stress on the expression and phosphorylation of endothelial nitric oxide (NO)-synthase was monitored, but endothelin 1 levels were found to be increased, contributing to the endothelial stiffening processes [13]. Furthermore, elevated MGO concentrations in aortic endothelial cells were associated with the increased expression of genes associated with coronary artery disease [14].

Animal data are available that show that the arterial response to acetylcholine stimulus was reduced after 24 weeks of diabetes. Immunohistology proved glyoxalase expression, Carboxymethyllysine (CML), and Hydroimidazolone accumulation in the mesenteric arteries of the animals as a consequence of AGE formation and accumulation. The overexpression of GLO1 in these animals could partially restore the acetylcholine response. GLO1 overexpression did not affect NO generation and endothelium-independent vasorelaxation, but the NO-independent relaxation was improved. The overexpression of GLO 1 reduced the levels of CML and Hydroimidazolone efficiently. The endothelial function parameters, such as ICAM and VCAM, were improved upon increased MGO detoxification [15]. If MGO is given to rats, diabetes-like microvascular changes were observed, including impaired vasodilation [16], a deleterious action on cutaneous vessels with endothelial cell loss, basement membrane thickening, and increased oxidative stress [17].

Concerning experiments in humans, results from several working groups support the direct effects of AGEs and concomitant elevated levels of MGO in cardiovascularly healthy patients with type 2 diabetes after a single meal rich in AGEs. Flow-mediated dilatation and reactive hyperaemia were more intensively affected following the AGE-rich meal compared to an isocaloric meal with a low AGE content, being steam-boiled instead of pan-fried. Moreover, after AGEs have been applied, patients presented with higher blood levels of MGO following the AGE-rich meal if compared to the low AGE meal, indicating exogenous AGEs as a potent source of MGO (production). In line with the clinical observed negative effect on vascular function, the levels of E-selectin were increased to 150% [18]. Diabetes mellitus is a potent driver of the MGO/AGE-derived vascular effects, and metabolically healthy people show less response to this stimulus. 

Thus, at every level—cellular, and living organism, including humans—MGO is identified as a toxic compound, shifting the balance to endothelial dysfunction and early atherosclerotic effects if the levels exceed certain limits (Figure 2).

## 4. The Mechanisms

The Brownlee hypothesis may serve as a basis for understanding the pathologic mechanism involved in endothelial dysfunction [19]. The endothelium relies on glycolysis for ATP production and uptaking glucose via GLUT1 [20]. Glucose might enter the polyol way and is metabolised via sorbitol into fructose. This process consumes the reduced glutathione (GSH) and thus reduces the antioxidative capacity. ROS production is detected as a consequence of this reaction, in addition to AGE production. At the level of the glycolysis intermediates, glyceraldehyd 3-phosphate (G3P) and dihydroxyacetone phosphate (DHAP), MGO might be produced via dephosphorylation. The pentose phosphate pathway and Krebs–(TCA) cycle mainly contribute to the pathophysiology via ROS generation by the reduced NAD(P)H availability or impairments in the respiratory chain [21]. The hexosamine pathway results in glycosylation of transcription factors responsible for the expression of TGFß-1 and PAI-I or VEGFR-2, resulting in impaired vascular and fibrinolytic function and neoangiogenesis. Finally, by increasing the concentration of diacylglycerol, proteinkinase C is activated, yielding increased vasoconstrictive compounds such as endothelin 1 and a decrease in vasodilative compounds such as eNOS. The unifying hypothesis considers mitochondrial ROS production as the final, upstream and harmful event [19] (Figure 3).

The extracellular stimulation of endothelial cells with MGO resulted in the generation of ROS. Peroxynitrite is formed from superoxide radicals resulting from eNOS uncoupling and related effects. The phosphorylation of p53, Akt, and JNK mediates apoptosis and stimulates the ROS activation of p50/p6- induced transcriptions of cyclooxygenase-2 (COX2) and inducible NO synthase (iNOS) [22]. The increased concentrations of MGO or MGO-H1 adducts do not directly inhibit eNOS activity, as was originally supposed from the structural homology of MGO-arginine adducts to asymmetric dimethyl arginine (ADMA) [23]. MGO decreased the SER1177 phosphorylation of eNOS with the consequence of eNOS uncoupling and the generation of superoxide anions and hydrogen peroxide [24]. eNOS uncoupling is a strong driver of endothelial dysfunction [25] and cardiovascular events such as hypertension, stroke, and heart failure [26]. Long-term exposure to MGO results in a decreased expression of eNOS [27]. Therefore, eNOS is targeted to MGO damage on the expression level, the phosphorylation grade, and by uncoupling.

Angiogenesis plays an important role in endothelial function. The hexosamine pathway induces VEGFR2 glycosylation that, in concert with Galectin-1, can provoke VEGF-independent signalling resulting in increased fibrosis. During angiogenesis, endothelial cells strongly depend on glycolysis and anaerobic glycolysis, yielding L-lactate. This, in turn, competes with α-ketoglutarate for pyruvate dehydrogenase-2 (PDH-2), leading to the stabilisation of hypoxia-inducible factor-1 α (HIF1α) and the upregulation of proangiogenic genes [21]. 

Apoptosis and autophagy are induced in MGO-treated aortic cells. Cyto-ID imaging identifies autophagic vacuoles. Autophagy is not only involved in endothelial cell survival and death but also modulates important endothelial cell functions, including homeostasis/thrombosis, angiogenesis, and NO production. Increased autophagy may be regarded as an alarm system for the cellular-stress response. During MGO treatment, autophagy mechanisms are increased via the ROS/MAPK pathway; they are fatal and lead to cell death via apoptosis. Anti-angiogenic mechanisms include the activation of the mTOR/Akt signalling pathway. Both pathways are involved in MGO-induced aortic endothelial cell dysfunction and, as a consequence, lead to late-diabetic complications [28].

## 5. Therapeutic Possibilities

As the endothelium is the central way to the organ, its dysfunction provokes atherosclerotic events, and thus the protection of the endothelium is organ protective [29]. Several substances have been investigated to overcome the negative effects of MGO exposure. These can be divided into scavengers such as aminoguanidine, pyridoxamine, L-carnosine, and others; secondly, those which inhibit formation, such as the well-known antidiabetic metformin, vitamin B1 such as Benfotiamin, angiotensin-converting enzyme inhibitors, angiotensin-receptor blockers, statins, ALT-711 (Alagebrium) and thiazolidinediones, or anti-obesity therapies; thirdly, the glyoxalase 1 inducers of which the combination of trans-Resveratrol (tRES) and Hesperitin (HES) is the most promising [9] (Figure 4).

The role of AGEs in disease and new compounds interfering with their effects is still under investigation in preclinical and clinical settings. Pyridoxamine has proven to be effective in an animal model as an antiglycating agent, protecting against high-fat diet (HFD)-induced body weight gain, hyperglycaemia, and hypercholesterolaemia [30]. In addition, pyridoxamine supplementation prevented impairments to glucose metabolism and insulin resistance in both HFD-induced diabetes and in db/db obese mice and inhibited the expansion of the adipose tissue and adipocyte hypertrophy. The beneficial effects on fatty liver content and vascular function upon pyridoxamine supplementation were demonstrated as well, highlighting the potential benefit of this substance in obesity and prediabetes [30]. Benfotiamin proved its beneficial effects in type 2 DM patients in regards to vascular function in terms of restoring reactive hyperaemia and laboratory parameters of endothelial function if given in advance of an AGE-rich meal [31]. In addition, Benfotiamin is a licensed drug and approved for the treatment of diabetic polyneuropathy (DPN) in several countries worldwide [32], and its use in symptomatic DPN is recommended [33]. Still, new substances are under clinical evaluation [34]. For a recent overview of new clinical substances, the reader is referred to Shen et al. [35].

The polyphenols, trans- Resveratrol and Hesperitin, work alone but are even more effective if applied in equimolar concentrations designed to prevent MGO-induced endothelial dysfunction in terms of the expression of VCAM, RAGE, E-selectin, and ICAM. The first clinical trial applying the combination to obese, prediabetic subjects proved its positive effects in preventing MGO-derived deleterious effects. Treatment with tRES/HES even delayed diabetes diagnosis in these subjects [36]. Thus, these substances will gain further attention in future research.

## 6. Role of MGO in Selected Vascular Diseases

### 6.1. Microvascular Disease

The most prominent microvascular diseases in diabetes are retinopathy, nephropathy, and peripheral neuropathy, and MGO contributes to the development and aggravation of all three.

Diabetic retinopathy (DR) is the most common diabetic microvascular disease, with more than one-third of patients with diabetes being affected [37]. Classical risk factors include diabetes duration, HbA1c, cholesterol levels, and blood pressure. DR is more common in type 1 diabetes mellitus and is associated with systemic vascular complications [38]. The formation of microaneurysms is a hallmark of the disease in its non-proliferative state and is associated with the proliferation of endothelial cells in addition to the loss of pericytes and structural alterations to the capillary basement membrane. Progressing to proliferative DR with aberrant neovascularization and retinal detachment is often associated with bleeding and pre-retinal or vitreous haemorrhage. Most commonly, macular oedema caused by retinal thickening contributes to vision loss in diabetic patients [39]. Patients with T1DM or T2DM with retinopathy present with higher serum values of MGO-H1, and the concentration depends on retinopathy status [40,41]. The retinal levels of MGO-H1 significantly increased in diabetic rats after a 24 weeks period of diabetes [42]. In cell culture, the overexpression of the detoxifying system, Glyoxalase 1, protected against the apoptosis of pericytes and endothelial cells in hyperglycaemia, suggesting the deleterious role of MGO in the early phases of retinopathy, mainly by apoptosis of pericytes [43]. Clinically, with ageing, the increased concentrations of MGO-derived AGEs contribute to the development of age-dependent cataracts, supporting the notion that GLO1 is strongly involved in the disease [44].

The progression of diabetic kidney disease in terms of nephropathy is the most common complication leading to dialysis in the world [45]. On a cellular basis, nephropathy develops as a consequence of a dysfunction in the endothelium of renal arterioles and the glomeruli, with a loss of interaction between the endothelial cells and glomerular podocytes. This is the basis for the clinical observation of hyperfiltration and increased permeability observed in microalbuminuria [46]. Microalbuminuria closely correlates with glycaemic control in diabetic patients, which is also a strong predictor of a progression to macroalbuminuria, and thus end-stage renal disease [47]. 

Again, the plasma levels of MGO and its derived AGEs positively correlate with eGFR (glomerular filtration rate) decline and albuminuria progression in both types of diabetes [48,49]. As AGEs and MGO are mainly cleared by renal excretion, eGFR decline and nephropathy per se increase their levels in the body. Decreased levels of GLO 1 were found to be drivers of albuminuria and a loss of function in endothelial and vascular cells, accompanied by an increase in oxidative stress [15]. Thus, diabetic nephropathy is not only driven by MGO, but by the reduced renal clearance of accumulated MGO and AGEs that exert their effect on the system itself.

As the third most common microvascular disease, diabetic polyneuropathy (DPN) accounts for the increased prevalence of chronic wounds with or without combined peripheral artery disease. Neuropathy in diabetes is clearly driven by MGO and AGEs. On the microvasculature level, the thickening of the vessel wall within the peripheral nerves, as well as the degeneration of pericytes and endothelial hyperplasia, are commonly observed [50]. Neuronal metabolism strongly depends on glycolysis, and glucose may enter the cell independent of insulin. Driven by hyperglycaemia, glucose may enter the neuron in an excessive amount, yielding an excessive MGO production. High MGO levels have been linked to an increased risk of developing DPN in T2DM [51], and higher levels have also been detected in T2DM patients suffering from DPN compared to those without DPN [52]. Nevertheless, studies suppose that the tissue contents of MGO are more relevant for the clinical phenotype than the plasma concentration [53]. This finding closely correlates with the aetiology of nerve fibre development and, finally, damage.

On the cellular level, Fukunaga et al. clearly demonstrated that Schwann cells (SCs) were directly targeted by MGO. MGO induced apoptosis in SCs in a dose-dependent manner, being accompanied by a reduction in intracellular glutathione concentration and the activation of the p38MAPK. The inhibition of the p38MAPK activation reduced MG-induced apoptosis in SCs. MGO scavengers, such as aminoguanidine and N-acetyl-L-cysteine, also prevented MGO-induced p38MAPK activation and apoptosis. In addition, intracellular glutathione content was restored following this treatment [54].

The therapeutic lowering of MGO by MGO scavengers resulted in a reduction in diabetes-induced hyperalgesia [52]. In addition to this, the transient receptor potential cation channel, subfamily A, member 1 (TRPA1), which was identified as an essential receptor in DPN, is modified by MGO at its cysteine residues, resulting in the reduced functionality of this receptor [55]. Again, reduced GLO1-levels have been identified as a potential mechanism for yielding increased MGO concentrations in animal experiments [56].

Taken together, MGO is involved in the aetiology of the three main microvasculature damages detected with diabetes mellitus and strategies to lower the amount of the toxic metabolites have been proven to diminish its effects. 

### 6.2. Macrovascular Disease

Without a doubt, atherosclerosis accounts for increased morbidity and mortality in the modern world, with cardiovascular disease (CVD) being the most prominent and fulminant form. For a long time, it was estimated that T2DM is associated with a more pronounced atherosclerotic risk, but it has become evident that both T1DM and T2DM share a comparable risk for the development of the disease. Without a doubt, glucose-lowering has improved patients’ outcomes in diabetes, and recent cardiovascular outcome trials have proven the beneficial effects mainly for glucagon, as with peptide 1 receptor agonists (GLP1RA), and sodium–glucose, as with transporters-2 (SGLT2), which may be additional to the glucose-lowering properties [57,58].

However, hyperglycaemia is proven to drive the atherosclerotic process, beginning with the development of endothelial dysfunction. The activation of the endothelium is provoked by diabetes and MGO that yield a pronounced expression of inflammatory molecules and adhesion factors [13,15]. The atherogenic LDL is targeted directly to MGO modification, resulting in modified LDL that, in addition to oxLDL, is an atherogenic compound [59,60]. Due to the modification, its affinity for the LDL receptor is reduced, yielding higher LDL plasma levels [59]. MGO-derived modifications of HDL result in a decreased antioxidative and anti-inflammatory capacity as well as a decreased half-life [61]. 

Dicarbonyl stress provokes adipose tissue dysfunction, which has been shown in experimental models [62]. For example, chronic administration of MGO (14 weeks, 50–75 mg/kg/day) to rats resulted in decreased irrigation of adipose tissue with clinical findings of hypoxia and macrophage infiltration in the glycated and fibrotic regions of adipose tissue [63]. This effect was directly attributable to MGO, as a high-fat diet (HFD) alone did not induce hypoxia. In addition, impaired adipose tissue blood flow contributed to the development of insulin resistance [63]. MGO supplementation in combination with HFD caused systemic metabolic dysregulation, indicated by higher free fatty acid levels, hyperinsulinaemia, glucose intolerance and insulin resistance [63]. MGO accumulation was previously reported to impair insulin-stimulated glucose uptake in both the adipose tissue of rats and 3T3-L1 adipocytes [64,65]. On the other hand, and obviously by different mechanisms, MGO treatment prolongs GLUT4 presence at the cell surface, allowing for massive and insulin-independent glucose influx in L6 myoblasts [66,67] and cardiomyoblasts (unpublished data by the author). By these mechanisms, MGO increases the underlying increased cardiovascular risk in diabetes by promoting gluco- and lipotoxicity.

GLO1 was identified in a large cohort genomic study as a major driver of cardiovascular disease (CVD), supporting the concept of glycation and reactive glucose metabolites as CVD-risk equivalent [14]. MGO-modified compounds have been detected in atherosclerotic plaques provoking further inflammation, and by the subsequent downregulation of GLO1, plaque ruptures occur that end up in clinically significant CVD endpoints [68].

In heart failure, mainly ischaemic heart failure, AGEs have been linked with severity and prognosis, and the accumulation of AGEs was correlated with the 1-year incidence of major cardiac events and death following myocardial infarction (MI) [69,70]. A mechanistic or causative role for AGEs in cardiovascular disease remains elusive. In fact, hyperglycaemia, hypoxia, ischaemia, inflammation, and oxidative stress drive the production of MGO and yield, in combination with GLO1 blockade, intra- and extracellular accumulation of MGO and MGO-derived AGEs (MGO-AGEs) [4]. Due to ischaemia, the infarcted heart undergoes a metabolic shift to derive ATP from anaerobic glycolysis, and inflammation, as well as oxidative stress, drive the formation of MGO [71,72].

Some years ago, Blackburn and colleagues sought to determine if this post-infarct-produced MGO has an effect on outcome after MI in mice in terms of adverse remodelling and cardiac dysfunction. They were able to confirm rapid MGO formation and accumulation after MI, mainly in type I collagen surrounding arterioles, which was reduced if GLO1 was overexpressed. Concomitantly, reduced MGO production was associated with reduced apoptosis, reduced scar size, and preserved cardiac function. In addition, higher vascular density due to improved neovascularization and less MGO-modified collagen that impair the angiogenic properties of peripheral blood mononuclear cells (PBMCs) were observed upon GLO1 overexpression. Thus, MGO does not only provoke MI by driving atherosclerotic events but impairs recovery after event independent of the infarct size [73]. MGO was shown to be an independent predictor of the prognosis of patients with congestive heart failure [69].

Very recently, Heber et al. found that up to a 2.4 fold increase in plasma MGO levels were detected 30 min after reperfusion by primary percutaneous coronary intervention after acute MI and that these levels further increased to 2.6 fold at 24 h after intervention. After 30 days, the levels reached the baseline values detected upon admission to the hospital. MGO values 24 h after intervention were negatively associated with outcome parameters such as the LVED values after 4 d. Therefore, the authors were able to explain an additional 23% of the variance in LVED on top of the 52% variance caused by myocardial necrosis and stretch [74]. 

In addition to atherosclerotic-derived cardiovascular events, heart failure (HF) contributes significantly to the increased morbidity and mortality among diabetic patients. Patients with diabetes present with 2.4 fold (men) to 5 fold (women) increased risk of heart failure [75]. Heart failure in diabetes is driven not only by coronary artery disease and hypertension but by metabolic disturbances provoking energy depletion and loss of function, named diabetic cardiomyopathy [76]. This entity is not yet fully understood, but it has been shown that MGO-overexpression prevented a hyperglycaemia-induced increase in circulating inflammatory markers and a loss of endothelial cell number. This was associated with less myocardial cell death, NO dimerization, RAGE expression, and inflammation. In vitro, TNFα in concert with MGO provoked an increased endothelial cell loss, clearly identifying MGO driven inflammation as a hallmark of diabetic cardiomyopathy [77]. 

More recent research revealed that myofilaments are targeted to MGO glycation, resulting in the contractile dysfunction of sarcomeres. Even type 2 diabetes mellitus patients without HF presented increased glycation of sarcomeric actin compared to non-diabetics, and this was correlated with decreased calcium sensitivity [78]. This clearly describes a direct link of MGO modification on heart function, an effect that was demonstrated before by showing that intercalated disk associated hsp27 is targeted to MGO modification in terminal heart failure [79].

Papadaki et al. showed that the MGO treatment of skinned human and mouse cardiomyocytes depressed both calcium sensitivity as well as maximal calcium-activated force dose-dependently. Myocytes derived from failing diabetic hearts were resistant to these myofilament functional changes, indicating immanent reduced function due to pre-existent MGO modification. Actin and myosin have been identified as MGO targets, and modifications depress calcium sensitivity and interaction with thin-filament regulatory proteins. Thus, myofilament modifications by MGO may account for a loss of function and the progression of heart failure, which is more pronounced in hyperglycaemia [80]. In L6 myoblasts, MGO treatment by intracellularly (via siRNA knockdown of GLO1) accumulated MGO or extracellularly administered MGO, an increased influx of glucose was detected, giving rise to the hypothesis that MGO induces glucose intoxication by prolonging the presence of GLUT4 on the cell surface; this yielded increased oxidative stress and apoptosis [66,67].

The increased risk for atherosclerotic events with diabetes mellitus includes a strong correlation with ischaemic stroke; patients with diabetes face an increased risk for stroke and vice versa [81,82]. T2DM is diagnosed three times more often in patients with ischaemic stroke than in controls [83], and the risk of stroke increases from 150% to 400% in patients with diabetes, with hyperglycaemia correlating directly with the risk of acute cerebrovascular incidents [81,84]. Diabetic patients are 2–6 times more susceptible to a stroke event, and this risk is amplified in younger individuals, hypertensive patients, or in those with complications in other vascular beds [85,86]. Less is known about the role of MGO during ischaemic stroke.

Vascular remodelling during hypertension development via vascular smooth cell proliferation is stimulated by MGO. Basilar smooth muscle cell (BASMC) proliferation was observed in a mouse model of angiotensin II (Ang II)-induced cerebrovascular remodelling during hypertension [87]. Mice presented with reduced GLO1 expression in basilar arteries, and this was negatively correlated with medial cross-sectional area and blood pressure in basilar arteries during hypertension. GLO1-knockdown increased, and GLO1 overexpression prevented Ang II-induced cell proliferation and cell cycle transition in BASMCs. In GLO1-overexpressing mice, cerebrovascular remodelling in basilar artery tissue during Ang II-induced hypertension development was improved, pointing to a role of GLO1 (and thus MGO reduction) as a negative regulator of hypertension-induced cerebrovascular remodelling [87]. These results were confirmed in a rat model where elevated MGO levels produced by vascular smooth muscle cells were shown to impair endothelial cell-mediated vasodilatation of cerebral microvessels, provoking cerebral ischaemia [88]. In human brain microvascular endothelial cells, MGO induced via the AGE/RAGE pathway ROS-mediated injury [89]. Increased ROS production was detected in human cerebral microvascular endothelial cells via ERK/JNK-signalling, reducing the blood-brain barrier integrity [90].

In summary, there is a clear contribution by MGO and the glyoxalase system in cerebrovascular remodelling, which comprise either the promotion of basilar smooth muscle cell proliferation in hypertension [87] or direct endothelial cell damage [87], resulting in endothelial and vascular dysfunctions [91].

Regarding peripheral artery disease (PAD), as the third of the big three macrovascular diseases, it is without a doubt that diabetes mellitus accounts for a significant increase in prevalence. Alone or in concert with diabetic polyneuropathy, PAD drives morbidity, disability, and mortality in affected patients in terms of developing diabetic foot syndrome. Again, the effect on arterial functioning and the inborn proatherosclerotic phenotype of MGO accounts for the increased risk of diabetic patients. Arterial stiffening is a measurable parameter for vascular functioning in the periphery. AGE deposition in the skin can be assessed by skin autofluorescence and is in correlation to AGE levels in the plasma. In a subset of patients of the European Prospective Investigation into Cancer and Nutrition (EPIC)-Potsdam cohort (EPIC-DZD), AGEs were measured independent of diabetes status and vascular stiffness. Skin autofluorescence was positively associated with markers of vascular stiffness, such as pulse wave velocity and augmentation index in diabetic patients, and the ankle-brachial index was inversely associated with skin autofluorescence across all sex, age, and glycaemia strata [92].

In patients with manifested PAD, measurements of skin autofluorescence clearly proved an increased level of AGEs in patients with diagnosed PAD, independent of cardiovascular risk factors and comorbidity. However, these conditions were associated with a further increase in AGE content if comorbidities were present [93].

Delayed wound healing is a prominent hallmark of diabetes, and diabetic foot is a common manifestation of late-diabetic complications driven by PAD and DPN. Upon the stimulation of macrophages with MGO and AGEs Bezold et al. could clearly show that glycation, but not AGE-modified serum proteins, activated macrophages to secrete proinflammatory cytokines such as interleukin 1β (IL-1β) and IL-8 and also affected IL-10 and TNF-α expression. As a consequence, the authors detected increased and persistent inflammation yielding in the chronification of wounds. In addition, glycation reduced efficiency, and this resulted in the impaired clearance rates of cellular debris and microbial load. Therefore, glycation contributes to disturbances in wound healing by the direct stimulation of macrophages and the altered expression of proinflammatory cytokines [94]. 

As with other microvascular diseases, MGO turned out to be a potent driver for disease onset and progression in any vascular bed and any organ affected. The strategies to reduce MGO content are associated with therapeutic success.

## 7. Conclusions

Without a doubt, hyperglycaemia aggravates atherosclerotic diseases, with endothelial dysfunction being a precursor of the disease on the micro- as well as macro- vascular levels. Vascular smooth muscle cells and platelet function are affected as well, and together with endothelial function, potentiate the risk.

Reactive glucose metabolites account for most of the hyperglycaemia-driven vascular and end-organ damages. As long-living modifications, AGEs have to be considered as “bad guys” provoking late-diabetic complications even after glucose-normalising interventions. Anti-MGO and anti-AGE therapies, in addition to glucose management, are, therefore, of growing importance in the successful treatment of diabetes and concomitant illness.

## Figures and Tables

**Figure 2 ijms-23-06186-f002:**
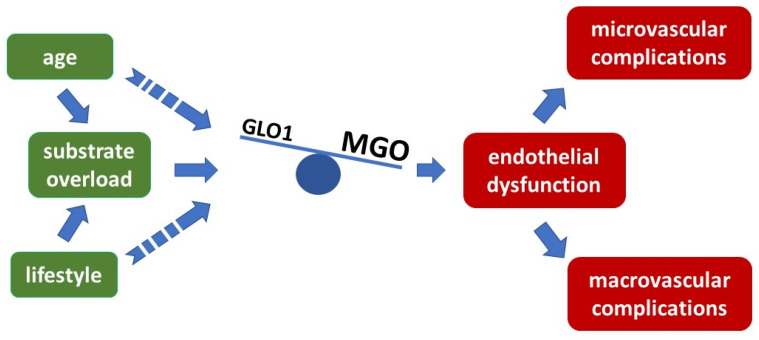
Imbalance of glyoxalase 1 activity and MGO concentration provoke endothelial dysfunction preceding atherosclerotic events, overflow with MGO exceeds the detoxifying capacity of GLO1 and results in endothelial dysfunction and further late complications.

**Figure 3 ijms-23-06186-f003:**
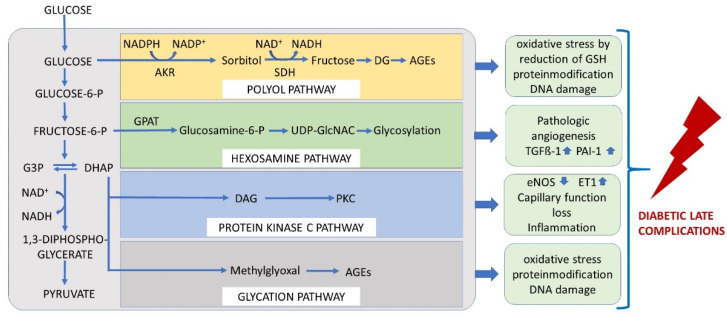
The unifying hypothesis comprises mechanisms by which glucose at each level of glycolysis contributes to the generation of late diabetic complications. Under physiological conditions, glucose is metabolised through the glycolytic pathway. Increase in intracellular glucose leads to increased flux of glucose to sorbitol via the polyol pathway, to an increase in the hexosamine pathway, to activation of protein kinase C (PKC), and to formation of advanced glycation end products (AGEs) in endothelial cells. AKR = aldose reductase, SDH = sorbitol dehydrogenase, GFAT = glutamine: fructose-6-phosphate amidotransferase, GlcNAc = N-Acetylglucosamin, DAG = diaclyglycerol, PKC = proteinkinase C, G3P = glyceraldehyd 3-phosphate, DHAP = dihydroxyacetone phosphate.

**Figure 4 ijms-23-06186-f004:**
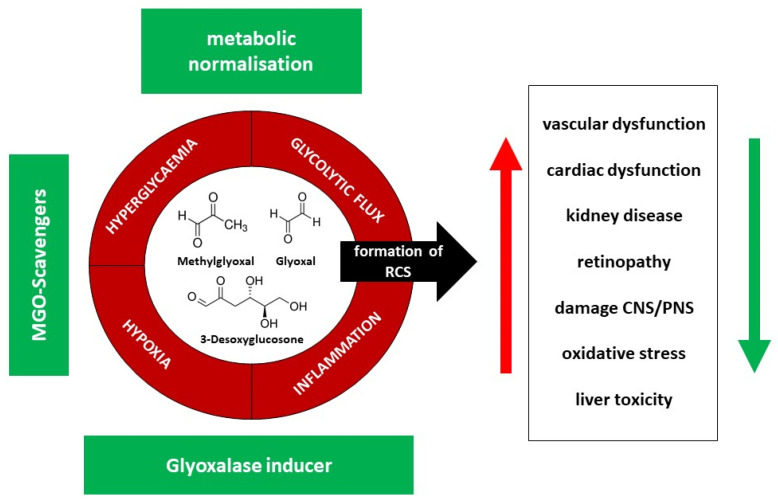
Effector pathways (red) and possible solutions (therapeutic options) (green) to overcome negative effects in reactive dicarbonyl and AGE generation, RCS = reactive carbonyl species, CNS = central nervous system, PNS = peripheral nervous system.

## Data Availability

Not applicable.

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
