# Peer review of "Dicarbonyl Stress in Diabetic Vascular Disease"

_ijms, 2022, doi:10.3390/ijms23116186_

Round 1

Reviewer 1 Report

The author reviewed the updated knowledge about the role of methylglyoxal (MGO) in diabetic endothelial dysfunction reported in the literature. MGO is a highly reactive dicarbonyl compound that is found at high levels in the plasma of diabetic patients. The topic is interesting dealing with relevant pathological aspects of MGO in vascular diabetes-related pathology.

To improve the quality of the manuscript we recommend:

Major comments

  • My overall opinion is that, while the topic is interesting, there is a lack of flow of information in some of the sections, especially in section 3 (The Observations) and section 5 (Therapeutic Possibilities) which are both poorly described. In this regard, please reorganise the study results presented in lines 89-97.
  • Currently, four potential mechanisms are described by which glucose and upstream intermediates of glycolysis induce damage to the endothelial cells and subsequently micro- and macrovascular complications - the sorbitol pathway, hexosamine pathway,  PKC pathway, and the glycation pathway. These are incompletely described in chapter 4. Supplementary, a brief presentation of the metabolism of the methylglyoxal could be a good factor to highlight this chapter.
  • No data presented related to the detoxification of MGO (major and minor pathways), such as the control of the Glo2, ALDH and AKR enzymes activity and the GLO1 gene expression (Schalkwijk and Stehouwer, 2020, https://doi.org/10.1152/physrev.00001.2019)
  • Fukunaga et al 2004, described the importance of MGO in diabetic neuropathy. In a murine model, MGO induces Schwann cells apoptosis through activation of p38MAPK. DOI: 10.1016/j.bbrc.2004.06.011. All these data should be added by the author in chapter 6
  • Sometimes the text is difficult to understand (kindly revised the text in lines 79-80 – ’’MGO values 24h after intervention were negatively associated with LVED values after 4 d.’’
  • What represents the red line in Figure 1?
  • There are no data in the main text (lines 156-161) related to Figure 2.
  • As a general observation, the figures would also benefit from typographical improvements to increase the readability of the text associated with the proposed diagrams.

Minor comments:

-     The author should explain all the abbreviations  (such as RCS, PNS, CNS) in the figure 2 legend

-     Please mention the complete name of CRP (line 64) in brackets at the beginning. The same observation for all abbreviations in the main text, such as PAI-1 in line 65, RCS in line 69, VACAM and ICAM in line 79.

-     In line 241 please replace  (/GLP1RA) with (GLP1RA)

-     English should be revised.

-     The references are written in different styles, especially the journal names.

Author Response

Answers to reviewers point are highlighted in italic.

Reviewer 1:

The author reviewed the updated knowledge about the role of methylglyoxal (MGO) in diabetic endothelial dysfunction reported in the literature. MGO is a highly reactive dicarbonyl compound that is found at high levels in the plasma of diabetic patients. The topic is interesting dealing with relevant pathological aspects of MGO in vascular diabetes-related pathology.

To improve the quality of the manuscript we recommend:

Major comments

My overall opinion is that, while the topic is interesting, there is a lack of flow of information in some of the sections, especially in section 3 (The Observations) and section 5 (Therapeutic Possibilities) which are both poorly described. In this regard, please reorganize the study results presented in lines 89-97.

Thank you for this comment, my plan was to start with cell culture experiments, and to go further to animals and finally to humans. The key animal experiments were not presented clearly and easy to read, thus reorganization - as proposed –  makes the results and information more easy to follow.

Currently, four potential mechanisms are described by which glucose and upstream intermediates of glycolysis induce damage to the endothelial cells and subsequently micro- and macrovascular complications - the sorbitol pathway, hexosamine pathway, PKC pathway, and the glycation pathway. These are incompletely described in chapter 4.

The Brownlee hypothesis is discussed in more detail now and a figure (figure 3) was added to underlie these basic principles.

Supplementary, a brief presentation of the metabolism of the methylglyoxal could be a good factor to highlight this chapter.

No data presented related to the detoxification of MGO (major and minor pathways), such as the control of the Glo2, ALDH and AKR enzymes activity and the GLO1 gene expression (Schalkwijk and Stehouwer, 2020, https://doi.org/10.1152/physrev.00001.2019)

The reviewer is absolutely right, the article would benefit from a “metabolism-chart” of MGO, which is now given as new figure 1 with short introduction in the text at the end of “clinical situation”. As ALDH and ARKs only represent minor MGO pathways these pathways are not discussed in more detail.As the focus of this articleshold be the vascular damage done by MGO, synthesis and degradation should be kept short…

Fukunaga et al 2004, described the importance of MGO in diabetic neuropathy. In a murine model, MGO induces Schwann cells apoptosis through activation of p38MAPK. DOI: 10.1016/j.bbrc.2004.06.011. All these data should be added by the author in chapter 6

Done, thank you for this advice, this brings results to the cellular level that have been missed in this chapter.

Sometimes the text is difficult to understand (kindly revised the text in lines 79-80 – ’’MGO values 24h after intervention were negatively associated with LVED values after 4 d.’’

I apologize for complicated sentence structure, this (and some other sentences) was corrected to an easier construct for better understanding.

What represents the red line in Figure 1?

Figure 1 which is now figure 2 has been changed, the red arrow was eliminated.

There are no data in the main text (lines 156-161) related to Figure 2.

The link to figure 2 which is now figure 4 has been shifted to above where it fits better to the text. Thank you for this comment.

As a general observation, the figures would also benefit from typographical improvements to increase the readability of the text associated with the proposed diagrams.

This has been worked on and legends have been improved – thank you.

Minor comments:

-     The author should explain all the abbreviations (such as RCS, PNS, CNS) in the figure 2 legend

-     Please mention the complete name of CRP (line 64) in brackets at the beginning. The same observation for all abbreviations in the main text, such as PAI-1 in line 65, RCS in line 69, VACAM and ICAM in line 79.

Thank you, abbreviations are now clarified upon first occurrence.

-     In line 241 please replace (/GLP1RA) with (GLP1RA)

Thank you for careful reading, corrected.

-     English should be revised.

Done, manuscript follows BE style.

-     The references are written in different styles, especially the journal names.

Harmonized to MDPI-style.

Reviewer 2 Report

This review presents the current knowledge regarding the effect of methylglyoxal (MGO) and AGEs accumulation on vascular function. The review is well-written and informative.

I have some minor comments to improve the review:

  1. The effect of MGO on inflammation and macrophage activity

Bezold, Veronika, et al. "Glycation of macrophages induces expression of pro-inflammatory cytokines and reduces phagocytic efficiency."Aging (Albany NY) 11.14 (2019): 5258.

  1. The accumulation of AGE and MGO impairs glucose utilization and might promote fatty acids synthesis and accumulation – possibly via SREBP1 and FASN. Adding the current knowledge about the effect of AGE and MGO on fatty acids synthesis and accumulation can improve the review.

For example, refer to:

Nigro, Cecilia, et al. "Dicarbonyl stress at the crossroads of healthy and unhealthy aging."Cell” 8.7 (2019): 749.

Dionne E. Maessen, Olaf Brouwers, Katrien H. Gaens, Kristiaan Wouters, Jack P. Cleutjens, Ben J. Janssen, Toshio Miyata, Coen D. Stehouwer, Casper G. Schalkwijk; Delayed Intervention With Pyridoxamine Improves Metabolic Function and Prevents Adipose Tissue Inflammation and Insulin Resistance in High-Fat Diet–Induced Obese Mice. Diabetes 2016; 65 (4): 956–966

  1. What therapies (besides regulating GLO1 levels) are currently in the clinic that directly target AGE (and specifically MGO) production?

I think its important to elaborate on current treatments and novel/under investigation drugs

For example, refer to:

Nenna, Antonio et al. “Pharmacologic Approaches Against Advanced Glycation End Products (AGEs) in Diabetic Cardiovascular Disease.” Research in cardiovascular medicine vol. 4,2 e26949. 23 May. 2015, doi:10.5812/cardiovascmed.4(2)2015.26949

  1. Page 6 line 241- “(/GLP1RA)”- please correct

Author Response

Answer to reviewers points in italic.

This review presents the current knowledge regarding the effect of methylglyoxal (MGO) and AGEs accumulation on vascular function. The review is well-written and informative.

I have some minor comments to improve the review:

  1. The effect of MGO on inflammation and macrophage activity

Bezold, Veronika, et al. "Glycation of macrophages induces expression of pro-inflammatory cytokines and reduces phagocytic efficiency."Aging (Albany NY) 11.14 (2019): 5258.

Thank you for highlighting this effect of MGO on macrophages and induction of inflammation. This was added to the review following the PAD chapter in macrovascular complications.

  1. The accumulation of AGE and MGO impairs glucose utilization and might promote fatty acids synthesis and accumulation – possibly via SREBP1 and FASN. Adding the current knowledge about the effect of AGE and MGO on fatty acids synthesis and accumulation can improve the review.

The reviewer is right to raise the point of MGO induced metabolic shifts, contributing to increased risk of vascular events in patients with diabetes. The metabolic impairment seen upon MGO supplementation or accumulation is now added in the chapter of macrovascular complications in the general introducing part citing relevant literature. The second suggested article can be found in the therapeutic options part of the review, as it describes the efficiency of pyridoxamine treatment.  

What therapies (besides regulating GLO1 levels) are currently in the clinic that directly target AGE (and specifically MGO) production?

I think its important to elaborate on current treatments and novel/under investigation drugs.

Although it is not the scope of this review to extensively describe therapeutic options the manuscript has been improved in this chapter by adding information of different treatment route like benfotiamine treatment in studies on vascular function and DPN and pyridoxamine in the model suggested by the reviewer. As this topic would have the potential of a review itself the reader is guided to a current available review on therapeutic possibilities, that to the authors point of view is actual and informative.

  1. Page 6 line 241- “(/GLP1RA)”- please correct

Done, thank you for careful reading.

Round 2

Reviewer 1 Report

The manuscript has been improved by responding to suggestions.

Some reservations remain about the Figures, which remain graphically unattractive and could be improved before publication, especially the font size.

Author Response

Dear reviewer,

thank you again for your valuable comments on the manuscript. The figures have been improved in terms of font size, more detailed information in the legends concerning abbreviations used. Figure 4 has been moved some lines above where it has been referenced to. 

With this revision some remaining spelling errors have been removed and the manuscript has been readily adapted to BE-spelling.